# Pharmacokinetic Profile Evaluation of Novel Combretastatin Derivative, LASSBio-1920, as a Promising Colorectal Anticancer Agent

**DOI:** 10.3390/pharmaceutics15041282

**Published:** 2023-04-19

**Authors:** Celina de Jesus Guimarães, Teiliane Rodrigues Carneiro, Marisa Jadna Silva Frederico, Guilherme G. C. de Carvalho, Matthew Little, Valder N. Freire, Victor L. B. França, Daniel Nascimento do Amaral, Jéssica de Siqueira Guedes, Eliezer J. Barreiro, Lídia Moreira Lima, Francisco W. A. Barros-Nepomuceno, Claudia Pessoa

**Affiliations:** 1Department of Physiology and Pharmacology, Drug Research and Development Center, Federal University of Ceara (UFC), Fortaleza 60430-275, CE, Brazil; 2Pharmacy Sector, Oncology Control Foundation of the State of Amazonas (FCECON), Manaus 69040-010, AM, Brazil; 3Department of Physics, Federal University of Ceara (UFC), Fortaleza 60440-900, CE, Brazil; 4Laboratório de Avaliação e Síntese de Substâncias Bioativas (LASSBio), Institute of Biomedical Sciences, Federal University of Rio de Janeiro (UFRJ), Rio de Janeiro 21941-590, RJ, Brazil; 5Institute of Health Sciences, University for International Integration of the Afro-Brazilian Lusophony, Redenção 62790-000, CE, Brazil

**Keywords:** combretastatin, PAMPA, pharmacokinetic, EGFR binding, tubulin

## Abstract

LASSBio-1920 was synthesized due to the poor solubility of its natural precursor, combretastatin A4 (CA4). The cytotoxic potential of the compound against human colorectal cancer cells (HCT-116) and non-small cell lung cancer cells (PC-9) was evaluated, yielding IC_50_ values of 0.06 and 0.07 μM, respectively. Its mechanism of action was analyzed by microscopy and flow cytometry, where LASSBio-1920 was found to induce apoptosis. Molecular docking simulations and the enzymatic inhibition study with wild-type (wt) EGFR indicated enzyme-substrate interactions similar to other tyrosine kinase inhibitors. We suggest that LASSBio-1920 is metabolized by O-demethylation and NADPH generation. LASSBio-1920 demonstrated excellent absorption in the gastrointestinal tract and high central nervous system (CNS) permeability. The pharmacokinetic parameters obtained by predictions indicated that the compound presents zero-order kinetics and, in a human module simulation, accumulates in the liver, heart, gut, and spleen. The pharmacokinetic parameters obtained will serve as the basis to initiate in vivo studies regarding LASSBio-1920’s antitumor potential.

## 1. Introduction

Pharmacokinetics is recognized as a crucial component of novel drug research and development (R&D). Many candidate molecules are eliminated during the early stages of clinical studies because they show unsuitable pharmacokinetic properties [1]. Associated with pharmacological effects, properties such as absorption, distribution, metabolism, and elimination are crucial to a drug’s advancement in research stages. Thus, extensive efforts have been made to propose tests that can evaluate them, even in the early stages of R&D [2].

Combretastatin A-4 (CA4) is obtained from the bark of *Combretum caffrum* (South African willow shrub) and binds to tubulin to destabilize microtubules and prevent cell division [3,4]. Based on this mechanism of action, CA4 arrests the cell cycle in metaphase and triggers apoptosis, resulting in a high toxicity profile against different human cancer cell lines [5]. CA4 is also a potent angiogenesis inhibitor with potential activity in treating metastatic cancer [6]. However, its clinical applications have been limited due to its low water solubility, low bioavailability, rapid metabolism, and systemic elimination [5,6,7]. These CA4 limitations encourage the search for analogues with a better biopharmaceutical profile [8].

The compound LASSBio-1586, a precursor to a series of synthesized analogues of CA4, stood out as highly cytotoxic to MDA-MB-435 cells [9]. From these data, new isosteres of LASSBio-1586 were synthesized, and LASSBio-1920 (Figure 1) was selected as the most potent against HL-60, OVCAR-8, and HCT-8 tumor cell lines. Docking studies suggested that LASSBio-1920 may be able to interact with β-tubulin through residues of DAMA-colchicine. Therefore, various potential chemical groups were bound to the molecule to achieve a strong interaction with the epidermal growth factor receptor (EGFR), a promising target for novel anticancer drugs [9,10,11,12]. However, the pharmacokinetic properties of LASSBio-1920 and if it can act using other pathways, such as inhibition of EGF receptor and/or activation of protein kinases, are unknown.

In this work, we evaluated the in vitro cytotoxicity and the LASSBio-1920’s action mechanism in human colorectal cancer cells (HCT-116) and non-small cell lung cancer cells (PC-9: overexpressed EGFR with L858R mutation). We also performed a preliminary pharmacokinetic study using in silico tools and suitable preclinical tests.

## 2. Materials and Methods

### 2.1. Materials

Dimethyl sulfoxide (DMSO), 99.5% (ACS reagent, Merck-Sigma-Aldrich, St. Louis, MO, USA); Methanol, Suitable for HPLC, gradient grade, ≥99.9%, (Merck-Sigma-Aldrich, St. Louis, MO, USA); Trifluoroacetic acid, UV/VIS ≥ 99.0% (suitable for HPLC, Merck-Sigma-Aldrich, St. Louis, MO, USA); 3-(4,5-Dimethyl-2-thiazolyl)-2,5-diphenyl-2H-tetrazolium bromide (MTT), 98% (Merck-Sigma-Aldrich, St. Louis, MO, USA); Panoptic kit (Laborlin, Pinhais, PR, Brazil); Propidium iodide, ≥94.0% (Merck-Sigma-Aldrich, St. Louis, MO, USA); RNase (Invitrogen^®^, Waltham, MA, USA); Sodium Citrate, (Merck-Sigma-Aldrich, St. Louis, MO, USA); Triton X-100, 1% (Invitrogen^®^, Waltham, MA, USA); Annexin V-PE/7-AAD Apoptosis Kit, (Elabscience^®^, Houston, TX, USA); Kinase-Glo Kit (Promega^®^, Madison, WI, USA); Phosphate Buffer Solution, 1.0 M, pH 7.4 (Merck-Sigma-Aldrich, St. Louis, MO, USA); Hydrochloric acid, ACS reagent, 37%, Merck-Sigma-Aldrich, St. Louis, MO, USA); Ethyl acetate, suitable for HPLC, ≥99.8% (Merck-Sigma-Aldrich, St. Louis, MO, USA); Sodium sulfate, ACS reagent, ≥99.0%, anhydrous (Merck-Sigma-Aldrich, St. Louis, MO, USA); Acetonitrile, suitable for HPLC, gradient grade, ≥99.9% (Merck-Sigma-Aldrich, St. Louis, MO, USA); Sodium hydroxide (Êxodo Científica, Sumare, SP, Brazil); 4-Acetylbiphenyl, 98% (Merck-Sigma-Aldrich, St. Louis, MO, USA); Tris-HCI Buffer (Invitrogen, Waltham, MA, USA); Sucrose (Dyslab, Curitiba, PR, Brazil); Bicinchoninic Acid Protein Assay Kit (Pierce, ThermoFisher Scientific, Waltham, MA, USA); Positive controls: LASSBio-1586, obtained in the Laboratório de Avaliação e Síntese de Substâncias Bioativas (LASSBio^®^, Rio de Janeiro, RJ, Brazil); Pelitinib, ≥98% (Merck-Sigma-Aldrich, St. Louis, MO, USA); Sunitinib (Pfizer, New York, NY, USA); Combretastatin (CA4), ≥98% (Merck-Sigma-Aldrich, St. Louis, MO, USA).

### 2.2. Cell Lines 

The human cancer cell lines with a regular EGFR expression (HCT-116, colon carcinoma; SF-295, glioblastoma; and PC-3, prostate adenocarcinoma) were obtained from the National Cancer Institute, Bethesda, MD, USA. The human cancer cell lines with overexpression of EGFR (H-1975, non-small cell lung adenocarcinoma; PC-9, lung adenocarcinoma; and N-87, gastric carcinoma) were purchased from ATCC and deposited in the Rio de Janeiro Cell Bank. The cell lines were maintained in RPMI-1640 medium supplemented with 10% fetal bovine serum, 2 mM glutamine, 100 U/mL penicillin, and 100 µg/mL streptomycin, at 37 °C, with 5% CO_2,_ and humidification.

Heparinized blood was collected from healthy, non-smoking donors who had not taken any drugs during the 15 days prior to sampling. Human peripheral blood mononuclear cells (HPBMC) were isolated by a standard density-gradient centrifugation method over Ficoll-Hypaque^®^ media (Merck-Sigma-Aldrich, St. Louis, MO, USA). HPBMC were washed and resuspended in RPMI-1640 medium supplemented with 20% fetal bovine serum, 2 mM glutamine, 100 U/mL penicillin, and 100 µg/mL streptomycin at 37 °C with 5% CO_2_. Phytohemagglutinin (3%) was added at the beginning of the culture.

### 2.3. Synthesis of LASSBio-1920

LASSBio-1920 was obtained from a condensation reaction according to the previously published procedure [11,12]. A solution of methyl benzoate (5.0 g, 22.1 mmol) was prepared in absolute ethanol (60 mL) and 64% hydrazine hydrate (21.4 mL, 442 mmol). The reaction mixture was refluxed for 2 h, and the resulting precipitate, 3,4,5-trimethoxybenzohydrazide (reagent 1), was filtered and obtained as a white solid, 92% yield, melting point range (m.p.) of 166–168 °C. Then, (E)-N′-(Benzo[b]thiophen-2-ylmethylene)-3,4,5-trimethoxybenzohydrazide (LASSBio-1920) was obtained as a white amorphous solid by condensation of the reagent 1 with benzo[b]thiophene-2-carboxaldehyde, m.p. 222.8 °C, resulting in an 84% yield [11,12].

### 2.4. Study of the In Vitro Antitumor Effect

#### 2.4.1. Cytotoxicity

The cytotoxicity of the compounds was determined using the MTT assay [13]. Cells were seeded in 96-well plates as follows: 0.7 × 10^5^ cells/well for adherent cancer cells, 0.3 × 10^5^ cells/well for suspended cancer cells, and 0.2 × 10^5^ cells/well for HPBMCs. The treatments were dissolved in DMSO and added to each well using an automated liquid handler for high-throughput screening (Biomek 3000—Beckman Coulter, Inc., Fullerton, CA, USA) and incubated for either 24, 48, and 72 h. The control group was treated with the vehicle. The final concentration of DMSO in the culture media was kept constant and below 0.5%.CA4 (0.07–8.6 μM), pelitinib (0.07–8.6 μM), both purchased from Sigma, sunitinib (0.07–8.6 μM), purchased from Pfizer, and LASSBio-1586 (0.07–8.6 μM) were used as positive controls. After incubation, the supernatant was decanted and replaced with fresh medium containing MTT (0.5 mg/mL). Three hours later, the MTT’s formazan product was dissolved in 150 µL of pure DMSO, and absorbance was measured at 595 nm (DTX 880 Multimode Detector, Beckman Coulter, Inc., Pasadena, CA, USA).

#### 2.4.2. Mechanism of Cytotoxicity

To describe how LASSBio-1920 induced cytotoxicity in HCT-116 and PC-9 cells (chosen based on cytotoxicity screening), each starting cell concentration (0.7 × 10^5^ cells/well) was incubated with LASSBio-1920 for 48 h at the three different concentrations (1.5, 3.0, and 6.0 μM for HCT-116 cells; 0.25, 0.5, and 0.75 μM for PC-9 cells) selected based on the IC_50_ values obtained by the MTT assay [14]. The control group received the same amount of vehicle, of which the concentration was kept constant (below 0.5%). Cell treatments were completed in triplicate. CA4 (0.2 or 0.5 µM) was used as the positive control. Stock solutions of the compounds tested were dissolved in DMSO and diluted in the culture medium to obtain the desired concentrations.

Morphological analysis: Morphological features of the cells were assessed using a light microscope (Olympus, Tokyo, Japan). Cells were harvested, transferred to cytosine slides, and stained using a Quick Panoptic Kit (Laborlin, Brazil). Before the analysis, cells were fixed with methanol and counterstained with 0.1% xanthine and 0.1% thiazine.

Determination of cell viability and membrane integrity: Cell viability and cell membrane integrity analyses were measured by propidium iodide exclusion (5 µg/mL, Sigma Aldrich Co., St. Louis, MO, USA), using flow cytometry. The cells were then plated in 24-well plates and treated with a test compound. After cell harvesting, treated and untreated cells were incubated with propidium iodide for 5 min and protected from light. Fluorescence was measured using flow cytometry in (FACSVerse™, BD Biosciences, San Jose, CA, USA).

DNA fragmentation and cell cycle analysis: The cell’s DNA was assessed using propidium iodide DNA staining followed by flow cytometry. Briefly, cells were treated with LASSBio-1920, harvested, and incubated, protected from light, in a solution containing 5 µg/mL propidium iodide, 5 µg/mL RNase, 0.1% sodium citrate, and 0.1% Triton X-100 at room temperature for 40 min. DNA fragmentation and cell cycle profiles were obtained through cell counting and fluorescence flow cytometry. Data were analyzed using ModFit LT (Verity Software House, Inc., Topsham, ME, USA).

Apoptosis assay: Apoptosis was analyzed using Annexin V/7-AAD staining followed by flow cytometry. Cells were collected and stained using an Annexin V-PE/7-AAD apoptosis detection kit. Annexin V binds to phosphatidylserine (PS) in apoptosis cells as PS translocates from the inner to the outer leaflet of the cytoplasmic membrane. 7-AAD is a fluorescent intercalator of DNA impermeable to the cell, providing a reliable indication of membrane integrity. Double staining distinguishes between viable, early apoptotic, necrotic, and late apoptotic cells. The resulting fluorescence of annexin V-PE at 583 nm and 7-AAD at 680 nm of every sample was analyzed by flow cytometry. The 7-AAD-positive and annexin V-negative cells were considered necrotic. Annexin V-positive cells (7-AAD-positive and 7-AAD-negative cells) were classified as apoptotic. 7-AAD-negative and annexin V-negative cells were deemed viable.

#### 2.4.3. Molecular Docking

The compound was constructed and analyzed conformationally based on the force fields of molecular mechanics; Spartan 8.0 (Wavefunction Inc., Irvine, CA, USA; License number: DQAIR/HASPUSB) was used to perform the analysis. The most stable conformer was reoptimized following the AM1 semi-empirical molecular orbital method. Subsequently, it was saved as a file for site-specific docking studies into the gefitinib binding site of the EGFR wt crystallographic structure available in the Protein Data Bank with code 2ITY [15]. The GOLD 5.0.1 program (CCDC) carried out the docking assays by employing a genetic algorithm for docking flexible ligands into protein binding sites and ranking the resulting poses according to their scores determined by available scoring functions. Hydrogen atoms were added to the protein structure according to the tautomeric and ionized states inferred by the program. The crystallographic gefitinib structure was removed to perform docking studies with the ChemScore scoring function, which contains specific energy terms for hydrogen bonding and lipophilic interactions [16]. The Pymol program aided the data and docking pose analyses. License numbers: Pymol (8588); Gold (G/414/2006).

#### 2.4.4. Enzymatic Activity

Inhibition of EGFR assay: The inhibition of EGFR activity was determined using the LANCE^®^TR-FRET (time-resolved fluorescence energy transfer) test according to the protocol described by Weber et al. (1984) [16]. The reference compound (PD153035) was tested simultaneously with LASSBio-1920 (1 and 10 µM). The LASSBio-1920 concentrations were determined based on IC_50_ values and the compound’s aqueous solubility. The experiment was carried out in agreement with Eurofins^®^’s Standard Operating Procedure [17].

Inhibition of human CK-1δ protein kinase assay: Test compound (10 µL), CK-1δ enzyme (10 µL), and buffer solution (20 µL) were added to each well of a 96-well plate containing 0.1% casein as substrate and 4 µM ATP. The assay buffer solution constituents were 50 mM HEPES at pH 7.5, 0.01% Brij-35, 10 µM MgCl_2_, 1 mM EDTA, and 0.01% NaN_3_. The final concentrations of DMSO were limited to 1%. After 60 min of incubation at 30 °C, the reaction was stopped by the addition of 40 µL of Kinase-Glo^®^. The luminescence was measured after 10 min using a Fluostar Optima multimode reader (BMG Labtechnologies GmbH, Offenburg, Germany). First, LASSBio-1920 was tested at concentrations of 10 µM, and 5 µM of SB415826, a potent CK-1δ enzyme inhibitor, was employed as a positive control [18].

#### 2.4.5. Data Analysis

Data were presented as mean ± standard error of the mean (SEM) from three independent experiments performed in triplicate, calculated using Prism 8.0 (GraphPad/Intuitive Software for Science, San Diego, CA, USA). The IC_50_ of the compounds was obtained using a non-linear regression of the cytotoxicity curves. Statistical analyses for multiple comparisons were performed using one-way ANOVA followed by Tukey’s test, and *p*-values of less than 0.05 were considered statistically significant.

### 2.5. Study of the Pharmacokinetic Parameters

#### 2.5.1. In Silico Simulations

Considering pharmacokinetic parameters, in silico simulations were performed using Certara Simcyp^®^ (Software version 19, Sheffiel, South Yorkshire, UK) to evaluate the total plasma compound concentration over time of LASSBio-1920 in humans and rat modules. The simulation inputs for both human and rat models consisted of compound-specific physicochemical properties such as log(P), pKa, and molecular weight. The virtual human population (n = 100) was designed with 10 trials (n = 10/trial), 20–50 years old, with cancer, 50% female, fasted, and infusion doses of 18, 36, 60, and 90 mg/m² (for 10 min). Test doses were determined based on the same CA4’s amount used in clinical trials [19]. The virtual animal population was constructed with healthy rats weighing 170 g, fasted infused (IV route; for 10 min) with 0.7, 1, and 4 mg/kg, beyond oral doses of 10 and 30 mg/kg. The test doses were defined considering similar studies [20,21]. For both, the simulation module considered a pharmacokinetic analysis for 24 h evaluating the main pharmacokinetic parameters, such as maximum concentration (CMax), maximum time (TMax), area under the curve (AUC), total clearance (CL), blood/plasma concentration ratio (B/P), distribution volume (Vd), distribution volume at steady state (Vss), plasma half-life (t_1/2_), and tissue/plasma partition coefficients (Kp).

#### 2.5.2. Chemical Stability 

To evaluate the chemical stability, 0.01 mmol of the compound was added to 10 mL of phosphate buffer, pH 7.4, and 10% hydrochloric acid, pH 1.2, and the mixture was agitated in a water bath at 37 °C with the end times of 0, 30, 60, and 240 min [22,23]. After each reaction, the mixture was extracted with ethyl acetate (3 × 20 mL). The extract was then dried with anhydrous sodium sulfate, and the solvent evaporated away. The product was analyzed in acetonitrile by HPLC (acetonitrile/water 4:1). For the samples tested at pH 1.2, the pH was neutralized with 10% sodium hydroxide before extraction and HPLC analysis. At appropriate intervals, the samples were further examined by thin-layer chromatography on silica gel (KieselGel 60 F245 Merck), and the products were visualized with an ultraviolet lamp (254 and 365 nm).

#### 2.5.3. In Vitro Plasma Stability 

The in vitro stability of the test compounds was studied in rat plasma. Blood from different Wistar rats was centrifuged with heparin at 20 °C for 15 min at 2000 rpm to separate the plasma, which was diluted to 80% with 0.05 M phosphate-buffered saline (PBS, pH 7.4) at 37 °C [23]. The test compounds were added to 200 µL of preheated plasma solution to yield a final concentration of 100 μM. The final concentration of DMSO in the incubation mixture was 0.5%. The assays were performed with agitation in a water bath at 37 °C and conducted in triplicate. Samples were taken at 0, 30, 60, and 240 min. After each sampling time, 500 µL of cold methanol was added to the mixture immediately, followed by 500 µL of cold acetonitrile. The samples were mixed and centrifuged at room temperature for 15 min at 13,000 rpm. The supernatant was analyzed by HPLC (acetonitrile/water, 7:3, with 0.1% trifluoroacetic acid) using an internal standard (4-acetylbiphenyl—final concentration of 20 µM).

#### 2.5.4. Microsomal Stability 

Livers were obtained from male Wistar rats (250 and 300 g) from the Federal University of Ceara (UFC, Fortaleza, Brazil). The livers were maintained in an ice bath and washed with 3 volumes of 0.05 M Tris-HCl and 0.25 M sucrose, pH 7.4. They were then sliced and homogenized in a Potter–Elvehjem glass-Teflon homogenizer. The homogenates were centrifuged for 30 min at 900× *g* at 4 °C, and the supernatant fraction was centrifuged at 10,000× *g* for 1 h at 4 °C. The pellet was discarded, and the supernatant fraction was centrifuged at 100,000× *g* for 1 h at 4 °C. The microsomal fraction in the resulting pellet was recovered. The pellet was washed and resuspended twice in 0.05 M Tris-HCl, condensed into a pellet again by centrifugation for 1 h at 100,000× *g* at 4 °C and, finally, resuspended in the Tris-HCl buffer. Metabolic assays were carried out with frozen microsomes in Tris–HCl buffer, stored at –80 °C. The protein content of the microsomal fraction was determined using Sigma’s bicinchoninic acid assay as suggested by the manufacturer. The mixture was composed of 1.3 mM MgCl_2_, 0.4 mM NADP+, 3.5 mM glucose 6-phosphate, and 0.5 U/mL glucose 6-phosphate dehydrogenase in 0.1 M phosphate buffer, pH 7.4, containing 1.5 mM EDTA and the test compound (final concentration of 100 µM, from stock solutions in DMSO) with a final volume of 250 µL [24].

The final concentration of DMSO in the incubation mixture was 0.5%. After pre-equilibration of the mixture at 37 °C for approximately 5 min, appropriate volumes of microsomal fraction were added to yield a final protein concentration of 4 mg/L. The mixtures were incubated at 37 °C for 0, 160, and 120 min. At the end of the incubation, 500 µL of cold methanol was added, followed by 500 µL of cold acetonitrile. After mixing, samples were centrifuged at room temperature for 15 min at 13,000 rpm. The supernatant was analyzed by HPLC (acetonitrile/water 7:3 with 0.1% trifluoroacetic acid) using an internal standard (4-acetylbiphenyl—final concentration of 20 µM). The experiments were performed in triplicate. Three control incubations were performed: (1) without cofactors, (2) without microsomal fraction and (3) with the 0.5% DMSO vehicle only [24].

#### 2.5.5. Permeability 

The following assay was performed according to the method described by Venzke et al. [25]. The parameter was evaluated using a parallel artificial membrane permeability assay (PAMPA) to assess the proportion of the drug that can migrate across the blood–brain barrier (BBB-PAMPA) and the gastrointestinal tract (GIT-PAMPA). For this, two 96-well plates were placed one above the other, and between them, an artificial hydrophobic polyvinylidene fluoride (PVDF) membrane, which was impregnated with a lipid solution. The top plate works as the donor compartment and the bottom one as the receiver compartment.

Parallel artificial membrane permeation assay for gastrointestinal tract (GIT-PAMPA): 1% (*w*/*v*) phosphatidylcholine was mixed with dodecane as a lipid solution to impregnate the PVDF membrane between the donor and receiver plates to stimulate intestinal permeability. A stock solution of 10 mM LASSBio-1920 was prepared, and 250 µL of this was subsequently added to 4750 µL of phosphate-buffered saline (PBS) and stirred overnight (test solution). A total of 180 µL of the mixture of PBS and DMSO (*v*/*v* 95:5) was added to the recipient plate, and 180 µL of the test solution was added to the donor plate. Immediately and carefully, the donor plate was placed above the receiver plate and incubated for 5 h at room temperature. The solution captured on the receptor plate was removed and subsequently analyzed by a UV detector (SpectraMax 5^®^—Molecular Devices, Sao Jose, CA, USA). The experiments were performed in quadruplicate.

Parallel artificial membrane permeation assay for the blood–brain barrier (PAMPA-BBB): It was performed according to the previously reported [26] method established by Lu and coworkers [27].

## 3. Results

### 3.1. Study of the In Vitro Antitumor Effect

The IC_50_ and selectivity index (SI) values of LASSBio-1920 against a panel of human tumor cell lines (normal-expressing EGFR: HCT-116, SF-295 and PC-3; overexpressing EGFR: H-1975, PC-9 and N-87) and human nontumor cells (HPBMC) after 72 h of incubation are presented in Table 1. LASSBio-1920 showed high activity against all tumor cell lines with IC_50_ values ranging from 0.06 to 1.68 μM, being more active than its precursor LASSBio-1586 and other inhibitors of receptor tyrosine kinases, such as pelitinib and sunitinib. Among the tumor cell lines used, HCT-116 and PC-9 were more sensitive to LASSBio-1920 treatment, showing IC_50_ values of 0.06 and 0.07 μM, respectively. In comparison to PBMCs, LASSBio-1920 was more active against HCT-116 (SI = 2.3) and PC-9 (SI = 2) cells. Based on IC_50_ and SI values, HCT116 and PC-9 cells were selected for a mechanistic study of LASSBio-1920 to determine its cytotoxic effect. This section may be divided by subheadings. It should provide a concise and precise description of the experimental results, their interpretation, as well as the experimental conclusions that can be drawn.

The IC_50_ values of LASSBio-1920 against HCT-116 and PC-9 cell lines after 24 and 48 h are presented in Table 2. Against HCT-116 cells, LASSBio-1920 had no effect after 24 h (IC_50_ > 100 μM) and a strong effect after 48 h (IC_50_ = 2.7 μM). A similar profile was observed with PC-9 cells, where the IC_50_ reduced from 78.80 to 1.01 μM after 24 and 72 h of treatment, suggesting that the cytotoxic effect of LASSBio-1920 may be time dependent.

The following assays were performed using treatment with three different concentrations of LASSBio-1920 (1.5, 3.0 and 6.0 μM for HCT-116 cells; and 0.25, 0.5 and 0.75 μM for PC-9 cells) and 48 h as incubation time for both cell lines. The LASSBio-1920 tested concentrations were selected based on the IC_50_ values obtained from the MTT assay.

Then, qualitative morphological analyses of HCT-116 (Figure 2A) and PC-9 (Figure 3A) cells treated with LASSBio-1920 were performed using the Quick Panoptic Kit. The photomicrographs showed that LASSBio-1920 induced phenotypic changes characteristic of apoptosis (nuclear fragmentation, chromatin condensation, apoptotic bodies, and cell shrinkage), similar to CA4 used as the positive control (0.2 or 0.5 µM).

Flow cytometry using the annexin V assay also confirmed the induction of apoptosis (Figure 4). At 6 µM (Figure 4A), LASSBio-1920 induced a significant increase in the amount of apoptotic HCT-116 cells (12.62 ± 2.26%) when compared with the negative control (5.14 ± 0.61%). Against PC-9 cells, we also observed remarkable induction of apoptotic events (15.49 ± 1.28%) at 0.75 µM of treatment when compared with the negative control (2.40 ± 0.17%) (Figure 4B).

Considering the justification for LASSBio-1920 synthesis, we performed its molecular docking study with EGFR-wt crystallographic structure. As seen in Figure 5, LASSBio-1920 displayed a cation-pi interaction (Lys745-phenyl), a hydrogen bond with the Asp855 NH group (2.4 Å), and non-polar interactions established by its benzothiophene ring. Furthermore, LASSBio-1586 and LASSBio-1920 break the saline bond of the αC-helix (similar property as to lapatinib), but do not have a polar interaction with the adenine region, unlike gefitinib, which has a hydrogen bonding interaction with Met793.

LANCE TR-FRET assay was performed using LASSBio-1920 at two concentrations (1 and 10 µM) to verify the enzymatic inhibition of EFGR-wt. The test compound showed similar and weak inhibition (less than 20%; data not shown) at either concentration. Weak inhibitory activity (below 10%; data not shown) against human CK-1δ protein kinase was also observed after incubation with LASSBio-1920 at 10 µM.

### 3.2. Study of the Pharmacokinetic Parameters

The prediction evaluations were performed using in silico tools. The pharmacokinetic parameters of LASSBio-1920 after a single IV administration at different doses obtained from the human simulation module Simcyp^®^ are presented in Table 3, tissue partition coefficients (Kp) in the human simulation module are presented in Table 4, and the concentration–time profiles are illustrated in Figure 6. CMax increased with the increase in administered doses 0.33, 0.67, 1.11, and 1.67 for 18 36, 60, and 90 mg/m^2^, respectively. TMax was 0.2 h (or 12 min) and remained the same for all administered doses. The AUC, area under the curve, also increased with increasing doses. CL and B/P values for the administered doses were the same, 25.72 L/h and 0.916, respectively. Vss was 4 L/kg, and it was equal for all the doses, while Vd showed a slight variation of 81.5 for the lowest dose and 79.6 for the highest dose. The last parameter obtained was t_1/2_ 2.19 h for the lowest dose and 2.14 h for the highest dose.

The tissue plasma partition coefficients (Kp) (Table 4) that presented the highest values were the liver with 15.04, followed by the heart with a Kp value of 9.06, gut with 8.65, and spleen with 8.40. The tissues with lower Kp were the lung, brain, adipose tissue, and bones, with Kp values of 1.62, 1.79, 1.96, and 2.52, respectively.

The pharmacokinetic profiles obtained in the animal simulation module of Simcyp^®^ for LASSBio-1920 (Table 5) showed that CMax, just like in the human module, increased with dose, with both administration routes. The values for CMax were 4.14 µM at 10 mg/kg and 13.43 µM at 30 mg/kg for the oral route (OR); and 0.7 µM for 0.7 mg/kg, 1 µM for 1 mg/kg and 3.94 µM for 4 mg/kg for the intravenous route (IVR). TMax was the same for all doses with OR, 0.61 h (or 36.6 min); and 0.01 h (or 36 s) with IVR. AUC, like CMax, increased with dose using both administration routes (Figure 7). The CL value was the same for all doses with either administration route, 0.69 L/h or 11.51 mL/min by OR and 0.33 L/h or 5.50 mL/min for 0.7 and 1 mg/kg doses, and 0.34 L/h or 5.61 mL/min for 4 mg/kg dose with IVR. B/P was 0.78, indicating that the highest drug concentration was in the plasma, and Vss was 2.72 L/kg for all doses with either administration route. Finally, t_1/2_ was equal to 1.13 h for OR and to 0.96 h for IVR at any dose of simulated administration.

In contrast with the values obtained in the human simulation module, the animal simulation module showed the highest tissue–plasma partition coefficient (Table 6) in the kidney, 4.48, followed by the adipose tissue, with 3.97 and the lung, with 3.92. Tissues with lower Kp were muscle, bone, and liver, with values of 2.09, 1.36 and 1, respectively. 

Following the pharmacokinetic analysis, we also evaluated six parameters (chemical stability at pH 2 and 7.4, permeability for the blood–brain barrier (BBB) and gastrointestinal tract (GIT), in vitro plasma stability and microsomal stability) to support further in vivo pharmacokinetic and pharmacodynamic studies.

First, Figure 8 shows that LASSBio-1920 was stable at pH 7.4 at all time points and unstable at pH 2 (beginning at 30 min of the experiment), suggesting that test compound stability is pH dependent.

Second, GIT (Table 7) and BBB (Table 8) permeability assays were performed for LASSBio-1920, considering its possible oral and systemic use. Taken together, the data obtained indicate that the test compound will be able to cross the BBB and reach the CNS (Pe = 11.48). In addition, it has a high predictive absorption fraction (99.7%) for the gastrointestinal tract.

To predict drug metabolism by enzymes, present in the blood plasma, LASSBio-1920 was incubated in rat plasma for 2 h. As shown in Figure 9, the test compound was stable at all time points of the assay, revealing promising plasma stability.

Finally, the metabolic stability of LASSBio-1920 was evaluated to advance its development as an anticancer prototype. The results reveal that the test compound was significantly metabolized by the microsomal fraction only in the presence of an NADPH-generating system (cofactor), indicating the action of flavin monooxygenases and/or cytochrome P450 (Figure 10). The metabolic analysis conducted by HPLC/MS revealed the formation of four major metabolites (M1, M1′, M2 and M2′). The chromatographic peaks of the metabolites M1 and M1′ were at 354.92 *m*/*z* (M–H), compatible with an O-demethylation process at position 3, 4 or 5 of the trimethoxybenzyl moiety of the LASSBio-1920 prototype. Moreover, metabolites identified as M2 and M2′ had peaks of 384.95 *m*/*z* (M–H), compatible with an oxidation process in the benzothiophene ring, which may represent oxidation of heteroatoms (S-oxidation), generating a sulfoxide as a metabolite and/or some hydroxylation in the benzothiophene ring (Figure 11 and Figure 12).

## 4. Discussion

The study describes an N-acylhydrazone derivative of LASSBio-1586 [9], named LASSBio-1920, with potent in vitro cytotoxic activity, especially against HCT-116 cells (human colon cancer) and PC-9 cells (human lung cancer with overexpression of EGFR with L858R mutation). The study was based on QSAR (quantitative relationships between structure and activity), on in vitro analyses, and on molecular anchoring studies for the synthesis of the LASSBio-1920 analogue [10,11,12]. For that matter, molecular modeling for rEGFR in silico studies and the analysis of preliminary pharmacokinetic parameters to increase bioavailability were also carried out.

In a previous study, LASSBio-1920 showed potent cytotoxicity and a better cytotoxic selectivity index against HL-60 cells [10,11,12]. In another study, the compound LASSBio-1586, in turn, stood out for exhibiting high cytotoxic effects against several cancer cell lines at nanomolar values (with MDA-MB435 showing the lowest IC_50_ value, i.e., 64 nM), as well as the high cytotoxic selectivity index and microtubule polymerization inhibitory activity [28,29]. In this study, LASSBio-1920 showed greater in vitro antitumor activity than LASSBio-1586 and tyrosine kinase receptor inhibitors (pelitinib and sunitinib) when tested against cancer cell lines HCT-116 and PC-9. 

One of the main causes of the adverse effects of cancer drugs is a low selectivity for tumor cells, where normal cells are also affected [30]. Considering the limits of the assay, LASSBio-1920 showed a higher selectivity profile than CA4 but lower than LASSBio-1586 for HCT-116 cells. Using PC-9 cells, LASSBio-1920 demonstrated greater selectivity for the cancer cell lines than normal cells in comparison to the compounds LASSBio-1586 and CA4. LASSBio-1920 also presented a higher cytotoxicity than both pelitinib and sunitinib, which showed clinical results against cancers with tyrosine kinase overexpression. Similar results were found with a series of new 3-alkyl-1,5-diaryl-1H-pyrazoles synthesized as CA4 analogues and evaluated for antiproliferative activity against three human cancer cell lines [31]. Another study on CA4 called steroid 22, which includes a 3,4,5-trimethoxy moiety similar to the structure of LASSBio-1920, exhibited moderate cytotoxic activity in the MCF-7 lines (IC_50_ 7.5 µM) and MDA -MB231 (IC_50_ 5.5 µM), with 24 h incubation, when compared to CA4 (IC_50_ 0.01 µM and 0.11 µM, respectively) [32]. In addition to the 3,4,5-trimethoxy subunit, LASSBio-1920 also has in its structure the benzothiophene group, which seems to be essential to its cytotoxic effect [10,11,12].

LASSBio-1920 showed time-dependent antiproliferative activity in both HCT-116 and PC-9 cells. This result suggests that this compound can affect cell division. These data are also related to the previously reported antimitotic effect [10,11,12]. Similarly, in other studies, caffeic acid phenethyl ester showed antimitotic action that was concentration- and time-dependent in HCT-116 cells. Furthermore, this compound showed cell growth inhibition in the MTT assay, corresponding to morphological changes, membrane integrity, cell viability, and DNA fragmentation of HCT116 and PC-9 cells treated with LASSBio-1920 in histological analyses [33,34]. These results were similar to those of CA4 (0.2 or 0.5 µM) used as a positive control in flow cytometry. LASSBio-1920 reduced cell density in both cell lines tested. Corroborating these data, a CA4 analogue called 8A, which showed increased antimitotic activity, caused apoptosis in HCT-116, SMMC-7221 liver cancer and A2780 ovarian cancer cell lines [35]. LASSBio-1920 promoted DNA fragmentation and induced PS externalization, suggesting that its mechanism of action may involve cell death by apoptosis in HCT-116 and PC-9 cells. In addition, pelitinib and sunitinib, which act through apoptosis, have less cytotoxic activity than LASSBio-1920.

Anchorage studies with EGFR tyrosine kinase indicated that LASSBio-1586 and LASSBio-1920 present different modes of interaction compared to gefitinib at the EGFR protein site (PDB: 2ITY). However, these compounds break the αC-helix saline bond, similarly to lapatinib (trade name: Tykerb^®^), a potent and reversible inhibitor of EGFR1 and EGFR2, approved in Brazil for the treatment of breast cancer [36]. In addition, similarly to CA4, it showed inhibition of some protein kinases [37,38]. The results with enzyme activities were not promising, indicating weak activity at interaction sites with human CK-1δ protein kinase, characteristic targets for treatment in human colorectal cancer cells, in which LASSBio-1920 showed antimitotic activity in this study. On the other hand, the break in the saline binding of the αC-helix indicates inhibitory activity against the EGFR tyrosine kinase receptor [39], so it is possible to suggest that LASSBio-1920 has some effect on the mutated EGFR target.

Although CA4 is one of the most potent antimitotic agents, its low availability in vivo leads to a reduced affinity for the target protein and, consequently, to a loss of its cytotoxic activity [7,40,41]. In phase I metabolism, a study carried out with CA4 in the presence of rat microsomes demonstrated that one of its metabolites formed by O-demethylation is in position 4 of the phenyl ring para position [42], which allows us to suggest the same for LASSBio-1920. Concerning the second metabolite, it is not possible to define exactly the position of O-methylation, whether C-3 or C-5, in the same way that we were unable for LASSBio-1920 due to the structural similarity between the two positions [43]. In the oxidative metabolism of LASSBio-1920, the cited authors indicated that it was metabolized in the microsomal fraction only in the presence of an NADPH-generating system (cofactor), indicating the action of the mononuclear and/or cytochrome P450 phagocytic system, which may suggest protection of this compound from liver first pass metabolism if administered orally. 

In the CNS, only a small amount of LASSBio-1920 was absorbed. On the other hand, in the gastrointestinal tract, LASSBio-1920 showed excellent absorption, approximately 99.7%. The high absorption in the GIT favors its metabolism. In addition, LASSBio-1920 was unstable in an acid medium (HCl buffer, pH 2) and stable in a practically neutral medium (phosphate buffer, pH 7.4). As for the absorption of LASSBio-1920 in the CNS, associated with its IC_50_ of 0.11µM, promising results could be obtained in a brain cancer cell line (SF-295—glioblastoma). LASSBio-1920 does not display metabolically labile sites regarding the action of carboxylesterases (the main hydrolytic enzyme of plasma in rodents), so it was not metabolized by plasma esterases, exhibiting substantial stability.

The pharmacokinetic profiles obtained for LASSBio-1920 showed that CMax, the maximum concentration reached in plasma [44], increased with dose in both human and animal simulation modules. TMax, i.e., the time in which the drug reaches its maximum concentration [44], remained the same, demonstrating proportionality with the change in dose and administration route in both simulation modules. AUC, representing the true metric of the drug in the blood, also demonstrates bioavailability, and like CMax, it increased with dose in both modules. CL, or drug clearance, indicates the blood flow completely free of a drug per unit of time [45,46]. The CL value for all administered doses in the human simulation module was the same, 25.72 L/h, which indicates saturation. In other words, the value remained unchanged and neither increased nor decreased with any change in the administered dose, as shown in Table 1. This type of phenomenon indicates zero-order kinetics, where a constant amount of drug is eliminated per unit of time, and the rate of metabolism remains constant with time, even at high drug concentrations [45,46]. This phenomenon can result in dangerously elevated plasma concentrations, which can cause toxic effects [45]. The data obtained from CL in the animal simulation module seem to have the same characteristic. This is because there was only a slight change in the highest dose administered by IVR. Considering B/P, a parameter that indicates the blood/plasma concentration ratio [47], both simulation modules indicated that the highest concentration of the drug was in the plasma. This result suggests that the compound could have a significant binding affinity for plasma proteins, which should be investigated in further studies. Vd and Vss represent the volume of fluid needed to contain the total amount of drug absorbed in the body and the volume of distribution at a steady state, respectively [48]. Vss on the human simulation module remained the same for all administered doses, just as it does the Vss on the animal simulation module. While Vd on the human simulation module had a slight variation between the lowest and the highest dose, Vss was considered a high value [49]. The last parameter obtained was t1/2 of 2.19 h for the lowest dose and 2.14 h for the highest dose. These values indicate a rapid elimination of LASSBio1920 from the plasma in the human simulation module, but longer than the value obtained with the lowest dose and shorter than that obtained with the highest dose when compared to the values obtained in a combretastatin phase I clinical trial [18].

The tissue/plasma partition coefficients (Kp) are important drug-specific input parameters in PBPK models and are used to quantify the distribution of drugs between tissues and plasma under steady-state conditions, that is, when no drug transfer occurs between tissue and plasma [50]. The organs with the highest partition coefficients were the liver, heart, gut and spleen. It is suggested that the high value determined for the liver was due to the possibility of predominant hepatic metabolism. Whereas other values can provide meaningful data to predict possible toxic effects and/or for targeting treatments for tissues in which antiproliferative activity against corresponding tumor cell lines showed more promising results, such as HCT-116, HCT-8, and PC-9 [10,11]. In the animal simulation module, the highest tissue plasma partition coefficient was seen in the kidney, followed by adipose tissue and the lung. In a study by Zhang and coworkers using another analogue of combretastatin, C118, similar outcomes were obtained in rats, with the highest accumulated concentrations of the compound being found in the lung, kidney and heart [4]. These data may suggest a predominance of renal excretion in rats in the same way that You and coworkers suggested for the combretastatin 4-N phosphate from its pharmacokinetic profile evaluation using the UPLC MS/MS technique [51]. The values obtained in animals will be used to initiate in vivo studies with the drug. In addition, simulation results will be compared with the real ones, hoping that there will be minor variations.

## 5. Conclusions

LASSBio-1920 showed promising cytotoxic potential and high selectivity for colon cancer cells (HCT-116) and lung cancer cells overexpressing mutated EGFR (PC-9). The mechanism of action was a prominent induction of apoptosis, compromising membrane integrity, decreasing cell viability, and causing DNA fragmentation, along with some cytostatic effect causing cell cycle arrest in G2/M phase. Docking studies reveal interactions of LASSBio-1920 with EGFR similar to some interactions displayed by a commercially available tyrosine kinase inhibitor (lapatinib), which may indicate an additional mechanism of action. As for the pharmacokinetic parameters obtained in vitro, LASSBio-1920 is metabolized by the microsomal fraction only in the presence of an NADPH-generating system (cofactor), indicating the action of mixed-function oxidases and/or cytochrome P450, and has plasma stability. LASSBio-1920 is stable at pH 7.4, making gastric protection strategies (encapsulation) necessary, as it has been shown to be unstable at pH 2. Considering in silico predictions, it may exhibit zero-order kinetic behavior. With these characteristics, it is important to note that the data obtained will form the basis for future studies on the development of the LASSBio-1920 nanoformulation, as well as for further in vivo experimental steps to determine pharmacokinetic parameters and antitumor activity.

## Figures and Tables

**Figure 1 pharmaceutics-15-01282-f001:**
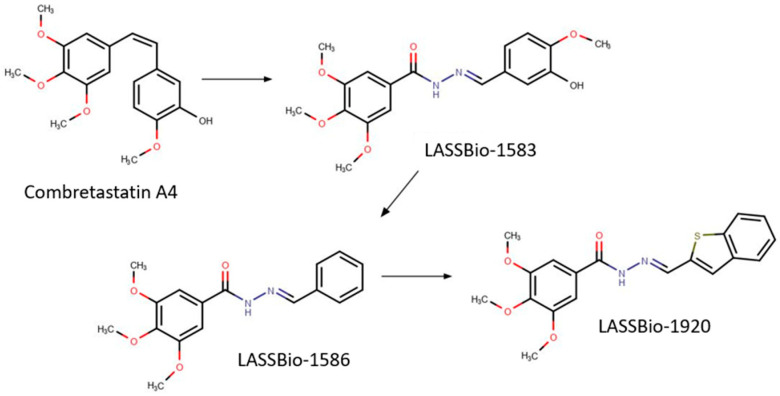
Structures of Combretastatin A4 and analogues LASSBio-1586 and LASSBio-1920 [10]. Designed by MarvinSketch 20.4 Software (Chemaxon, Budapest, HUN).

**Figure 2 pharmaceutics-15-01282-f002:**
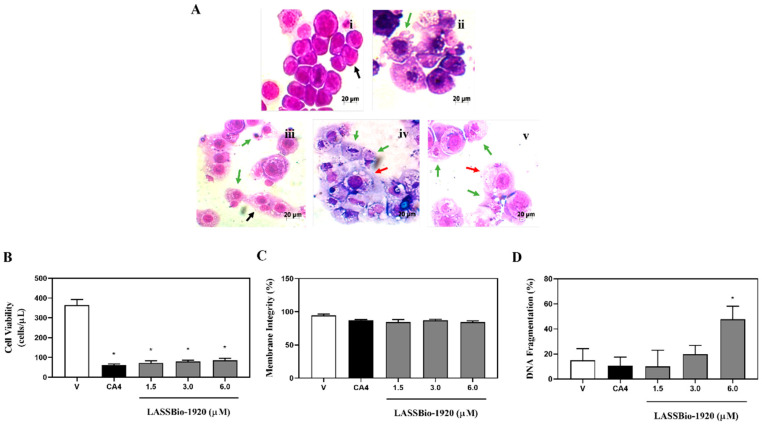
Morphological changes (**A**), cell viability (**B**), membrane integrity (**C**) and DNA fragmentation (**D**) of HCT116 human tumor cells treated with LASSBio-1920 (1.5, 3.0 and 6.0 μM) after 48 h of incubation. (**A**) (vehicle, i; positive control 0.2 µM, ii; LASSBio-1920 1.5 µM, iii; LASSBio-1920 3.0 µM, iv; and LASSBio-1920 6.0 µM, v) light microscopy (200×) analyses of stained cells with Quick Panoptic Kit (black arrows show mitotic cells, green arrows show apoptotic cells, and the red arrow shows cytoplasmic vesicle formation). (**B**–**D**), flow cytometry using propidium iodide. The control group received the same amount of vehicles whose concentration was kept constant (DMSO, 0.5% in medium). CA4 (combretastatin-A4; 0.2 or 0.5 µM) dissolved in DMSO was used as the positive control of the experiment. Data were presented as mean ± SEM from three independent experiments performed in triplicate calculated by Prism 6.0 (GraphPad/Intuitive Software for Science, San Diego, CA, USA). Statistical analyses for multiple comparisons were performed using one-way ANOVA followed by Tukey’s test. * *p* < 0.05 was considered statistically significant.

**Figure 3 pharmaceutics-15-01282-f003:**
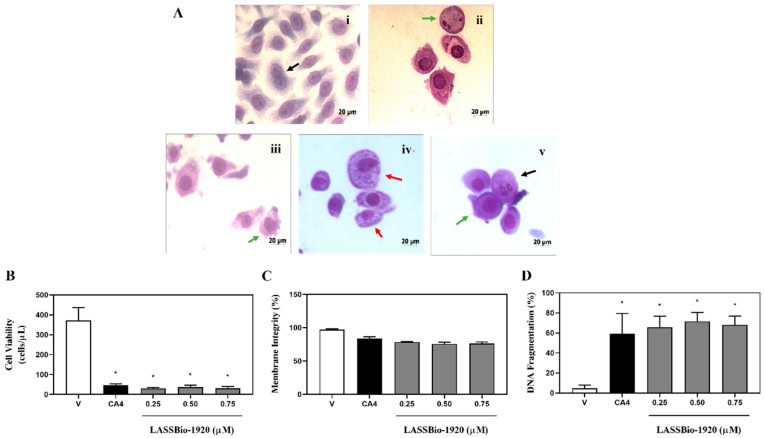
Morphological changes (**A**), cell viability (**B**), membrane integrity (**C**) and DNA fragmentation (**D**) of PC-9 cells treated with LASSBio-1920 (0.25, 0.5 and 0.75 μM) after 48 h of incubation. (**A**) (vehicle, i; positive control 0.2 µM, ii; LASSBio-1920 0.25 µM, iii; LASSBio-1920 0.5 µM, iv; and LASSBio-1920 0.75 µM, v), light microscopy (200×) analyses of stained cells with Quick Panoptic Kit (black arrows show mitotic cells, green arrows show apoptotic cells and red arrows show cytoplasmic vesicle formation). (**B**–**D**), flow cytometry using propidium iodide. The control group received the same amount of vehicles whose concentration was kept constant (DMSO, 0.5% in medium). CA4 (combretastatin-A4; 0.2 or 0.5 µM) dissolved in DMSO was used as positive control of the experiment. Data were presented as mean ± SEM from three independent experiments performed in triplicate calculated by Prism 6.0 (GraphPad/Intuitive Software for Science, San Diego, CA, USA). Statistical analyses for multiple comparisons were performed using one-way ANOVA followed by Tukey’s test. * *p* < 0.05 was considered statistically significant.

**Figure 4 pharmaceutics-15-01282-f004:**
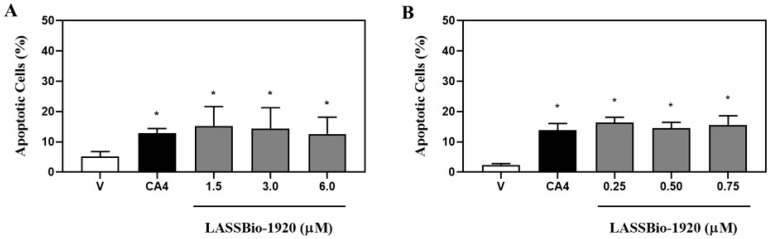
Apoptosis induction analysis of HCT-116 (**A**) and PC-9 (**B**) human tumor cells treated with LASSBio-1920 (1.5, 3.0 and 6.0 μM against HCT-116; and 0.25, 0.5 and 0.75 μM against PC-9) after 48 h of incubation. Control group received the same amount of vehicle whose concentration was kept constant (DMSO, 0.5% in medium). Cell data were presented as mean ± SEM from three independent experiments performed in triplicate calculated by Prism 6.0 (GraphPad/Intuitive Software for Science, San Diego, CA, USA). Statistical analyses for multiple comparisons were performed using one-way ANOVA followed by the Tukey’s test. * *p* < 0.05 was considered statistically significant.

**Figure 5 pharmaceutics-15-01282-f005:**
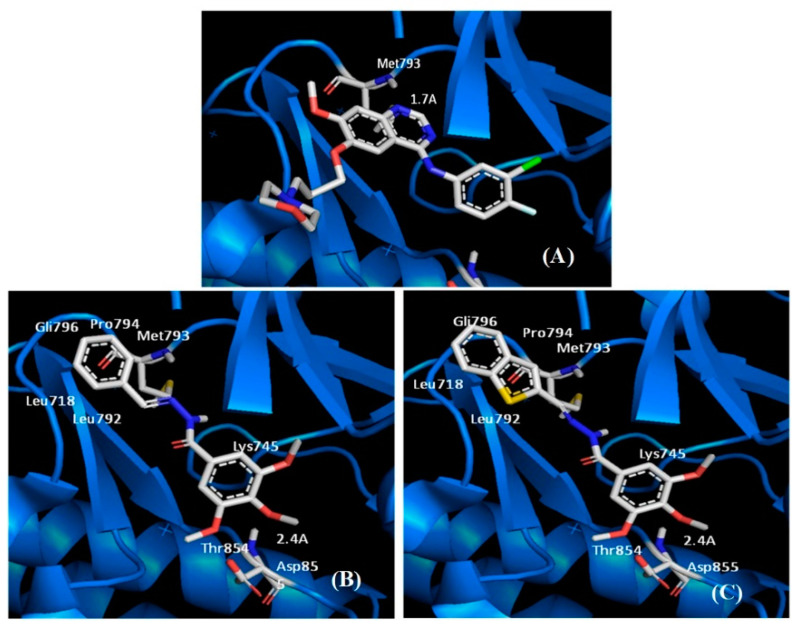
EGFR-wt binding site interactions. Carbon atoms are represented in grey, sulfur in yellow, oxygen in red, and nitrogen in blue. (**A**) Gefitinib on EGFR wt (PDB: 2ITY)—hydrogen bond interaction with Met793 (adenine region). (**B**) Interaction of LASSBio-1586 on EGFR wt (PDB: 2ITY). (**C**) Interaction of LASSBio-1920 on EGFR wt (PDB: 2ITY). Figures generated from the PyMol program.

**Figure 6 pharmaceutics-15-01282-f006:**
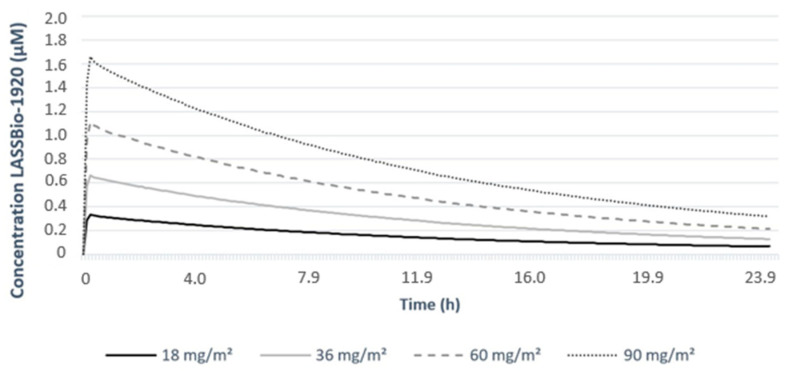
LASSBio–1920 concentration-time profiles in human simulation module.

**Figure 7 pharmaceutics-15-01282-f007:**
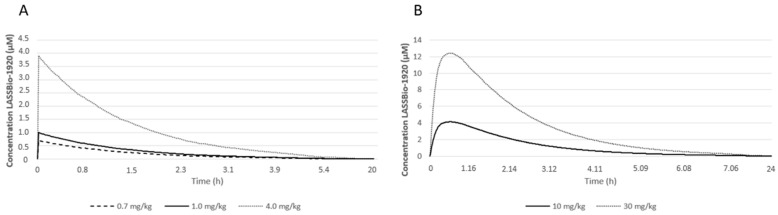
LASSBio-1920 pharmacokinetic profiles in animal simulation module for IV (**A**) and for oral (**B**) administration.

**Figure 8 pharmaceutics-15-01282-f008:**
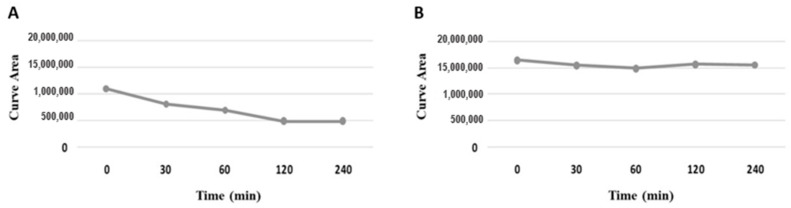
Chemical stability of LASSBio−1920 at pH 2 (**A**) and 7.4 (**B**).

**Figure 9 pharmaceutics-15-01282-f009:**
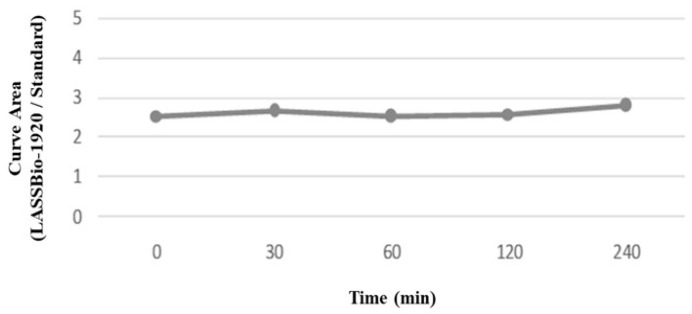
Plasma stability of LASSBio-1920 and the standard methyl biphenyl-4-carboxylate.

**Figure 10 pharmaceutics-15-01282-f010:**
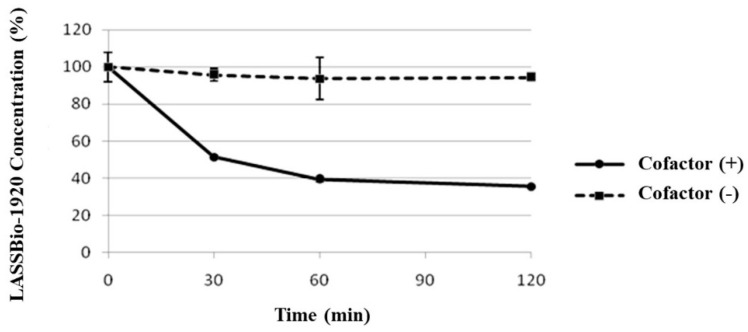
Metabolic stability of LASSBio-1920 using liver microsomal fraction of Wistar rats in the presence (+) or absence (−) of cofactor. Data are presented as a percentage of the compound concentration.

**Figure 11 pharmaceutics-15-01282-f011:**
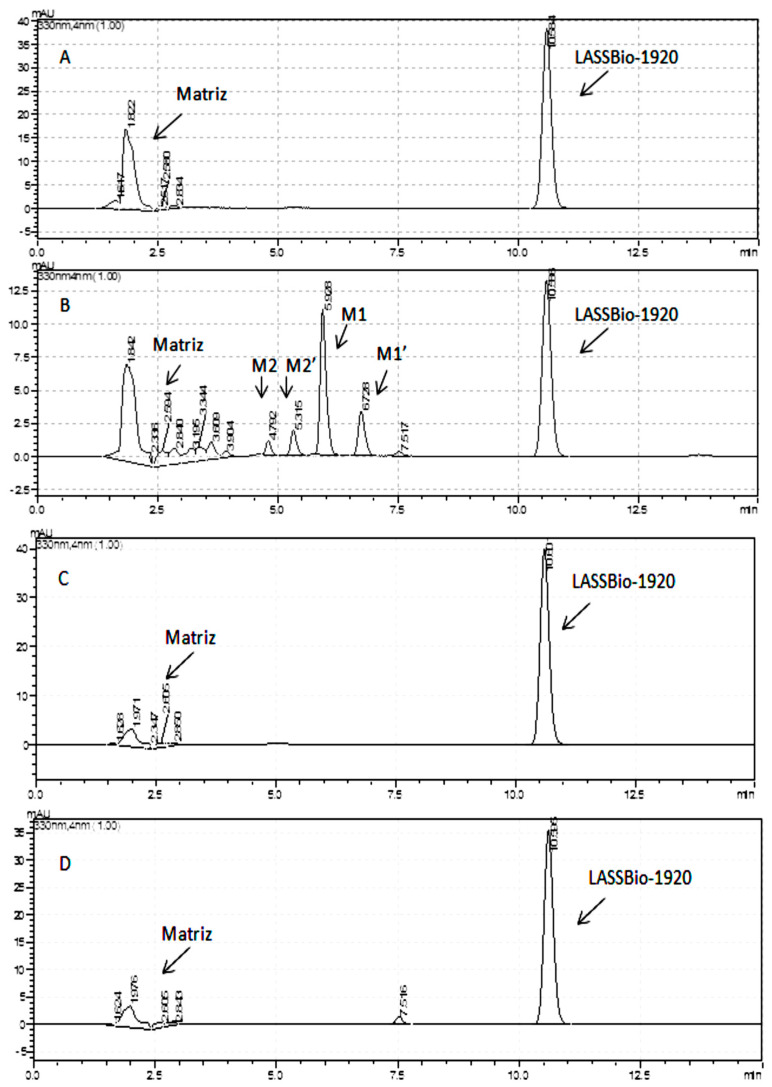
Metabolic analysis of LASSBio-1920 by HPLC/MS. Samples analyzed at 330 nm in triplicate: (**A**) LASSBio-1920, incubation time 0 min in the presence of NADPH−generating system; (**B**) LASSBio-1920, after 120 min of incubation in the presence of an NADPH−generating system; (**C**) LASSBio-1920, time 0 min in the absence of an NADPH generating−system; (**D**) LASSBio-1920, after 120 min of incubation in the absence of an NADPH−generating system. Mobile isocratic phase acetonitrile: water (1:1), 15 min and 1 mL/min.

**Figure 12 pharmaceutics-15-01282-f012:**
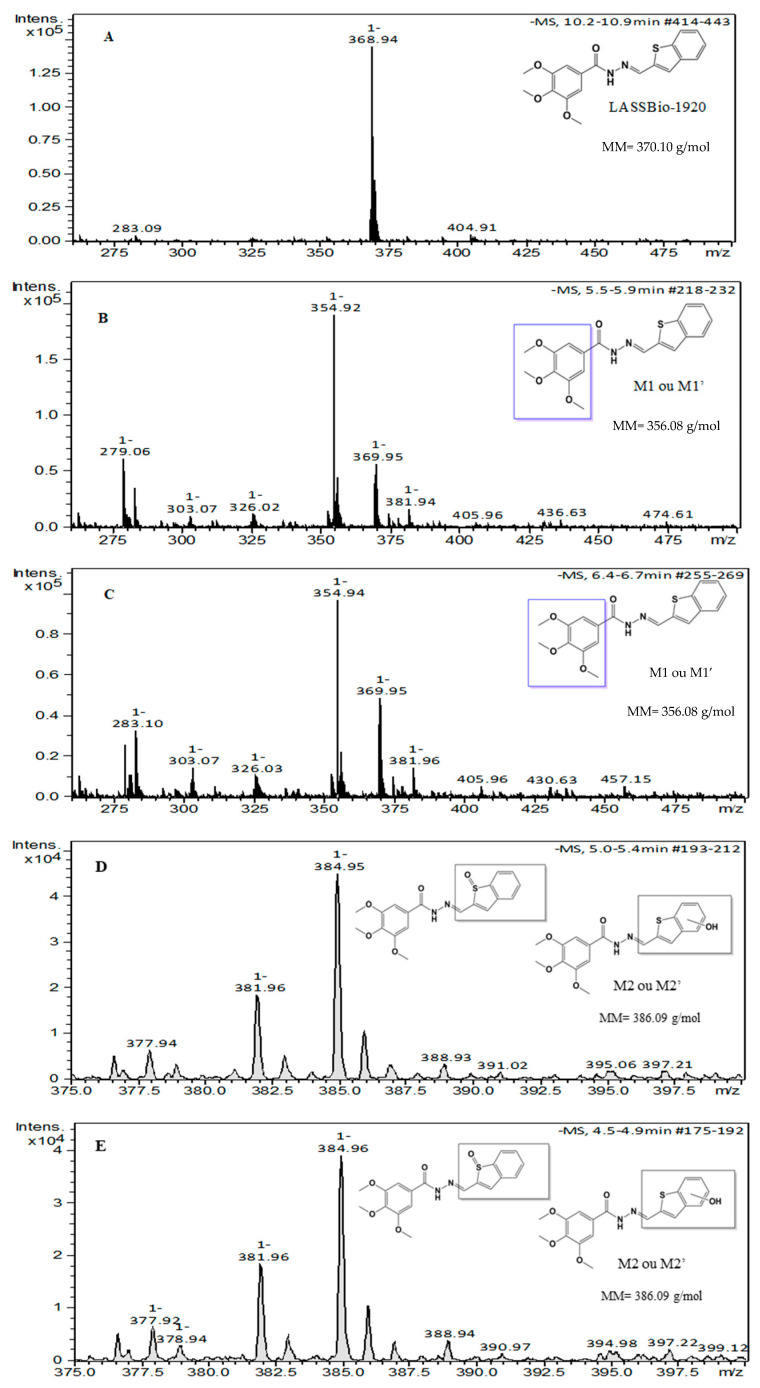
Mass spectrum (HPLC/MS) of LASSBio-1920 and its metabolites. (**A**) LASSBio-1920, 369 *m*/*z* (M–H); (**B**,**C**) Metabolites M1 or M1′, 355 *m*/*z* (M–H), compatible with the O-demethylation process that may occur at the 3, 4 or 5 of trimethoxybenzyl moiety; (**D**,**E**) metabolite M2 or M2′, 386 *m*/*z* (M–H), compatible with S-oxidation process and/or some hydroxylation in the benzothiophene ring.

**Table 1 pharmaceutics-15-01282-t001:** Cytotoxic effect of LASSBio-1920 at different concentrations (0.02–2.5 μM) against human tumor cell lines and human non-tumor cells after 72 h of incubation using MTT assay.

Compound	IC_50_ ^a^ μM (95% CI)							SI ^b^	
	HCT-116	SF-295	PC-3	H-1975	PC-9	N-87	HPBMC	HPBMC/HCT-116	HPBMC/PC-9
LASSBio-1920	0.06 (0.05–0.11)	0.11 (0.05–0.24)	1.68 (0.38–5.24)	0.21 (0.12–0.38)	0.07 0.045–0.10	0.44 (0.24–0.78)	0.14 (0.06–0.35)	2.3	2
LASSBio-1586 ^c^	0.32 (0.35–0.51)	0.26 (0.13–0.54)	0.92 (0.80–2.68)	2.83 (1.9–4.12)	0.75 (0.64–0.89)	0.82 (0.40–1.68)	1.34 (1.05–1.66)	4.2	1.8
CA4 ^c^	0.0021	0.008	0.008	0.035	0.11	3.34	0.0047	2.24	0.043
PELITINIB ^c^	ND *	ND *	ND *	4.4	0.53	0.33	ND *	ND *	ND *
SUNITINIB ^c^	ND *	ND *	ND *	4.48	20.8	10.9	ND *	ND *	ND *

^a^ Data are presented as half-maximal inhibitory concentration (IC_50_) in μM and 95% confidence intervals (95% CI) for human tumor cell lines with normal EGFR expression (HCT-116, colon carcinoma; SF-295, glioblastoma; PC-3, prostate adenocarcinoma), human tumor cell lines with overexpression of EGFR (H-1975, non-small cell lung adenocarcinoma; PC-9, lung adenocarcinoma; and N-87, gastric carcinoma), and human non-tumor cells (HPBMC, human peripheral blood mononuclear cells) obtained from three independent experiments performed in triplicate. IC_50_ was calculated using Prism 6.0 (GraphPad/Intuitive Software for Science, San Diego, CA, USA). Viability data were normalized to the control group (DMSO, 0.5% in medium). ^b^ Cytotoxic selectivity index (SI) represents IC_50_ for HPBMC/IC_50_ for the tumor cell line. ^c^ LASSBio-1586 (0.07–8.6 μM), CA4 (combretastatin-A4; 0.0006–10 μM), pelitinib (0.08–10 μM), and sunitinib (0.31–40 μM) were used as positive controls for the test. * Not determined.

**Table 2 pharmaceutics-15-01282-t002:** Cytotoxic effect of LASSBio-1920 at different concentrations (0.01–100 μM) against HCT-116 and PC-9 human tumor cells after 24, 48, and 72 h of incubation using MTT assay.

Compound	IC_50_ ^a^ μM (95% CI)					
	HCT-116			PC-9		
	24 h	48 h	72 h	24 h	48 h	72 h
LASSBio-1920	>100	2.7 (1.9–2.9)	0.06 (0.05–0.11)	78.80 (43.5–88.6)	1.01 (0.28–3.69)	0.07 (0.04–0.10)
CA4 ^b^	ND *	0.2	0.0021	2.60	1.38	0.11

^a^ Data are presented as half-maximal inhibitory concentration (IC_50_) in μM and 95% confidence intervals (95% CI) for HCT-116 (normal EGFR expression; colon carcinoma), and PC-9 (overexpression of EGFR) human tumor cells obtained from three independent experiments performed in triplicate. IC_50_ was calculated using Prism 6.0 (GraphPad/Intuitive Software for Science, San Diego, CA, USA). Viability data were normalized to the control group (DMSO, 0.5% in medium). ^b^ CA4 (Combretastatin-A4; 0.0006–10 μM) was used as positive control for the test. * Not determined.

**Table 3 pharmaceutics-15-01282-t003:** LASSBio-1920 pharmacokinetic profiles in human simulation module.

Parameter	Dose (mg/m²)			
	18	36	60	90
CMax (µM)	0.33	0.67	1.11	1.67
TMax (h)	0.20	0.20	0.20	0.20
AUC (µM.h)	3.74	7.48	12.47	18.70
CL (Dose/AUC) (L/h)	25.72	25.72	25.72	25.72
B/P	0.92	0.916	0.916	0.916
Vss (Subs)(L/kg)	4.00	4.00	4.00	4.00
Vd (L)	81.5	78.4	79.5	79.6
t_1/2_ (h) *	2.19	2.11	2.14	2.14

* Obtained by calculation formula: t_1/2_= 0.693; Vd/CL. CMax: maximum drug concentration; TMax: time to maximum concentration; AUC: area under the curve; CL: clearance; B/P: concentration ratio of the drug in the plasma and blood; Vss: volume of distribution at steady state; Vd: volume of distribution; t_1/2_: half-life time.

**Table 4 pharmaceutics-15-01282-t004:** LASSBio-1920 tissue partition coefficient (Kp) in human simulation module.

Tissue	Kp	Tissue	Kp
Liver	15.04	Pancreas	5.26
Heart	9.06	Skin	4.09
Gut	8.65	Bone	2.52
Spleen	8.40	Adipose	1.96
Muscle	7.49	Brain	1.79
Kidney	7.44	Lung	1.62

**Table 5 pharmaceutics-15-01282-t005:** LASSBio–1920 PKPD Profiles in animal simulation module.

Parameter	Dose (mg/kg)				
	Oral Route		Intravenous Route		
	10	30	0.7	1	4
CMax (µM)	4.14	12.43	0.70	1.00	3.94
TMax (h)	0.61	0.61	0.01	0.01	0.01
AUC (µM.h)	9.77	29.32	0.97	1.39	5.45
CL (Dose/AUC) (L/h)	0.69	0.69	0.33	0.33	0.34
CL (Dose/AUC) (mL/min)	11.51	11.51	5.50	5.50	5.61
B/P	0.78	0.78	0.78	0.78	0.78
Vss (Subs) (L/Kg)	2.72	2.72	2.72	2.72	2.72
t_1/2_ (h)	1.13	1.13	0.96	0.96	0.96

CMax: maximum drug concentration; TMax: time to maximum concentration; AUC: area under curve; CL: clearance; B/P: concentration ratio of the drug in the plasma and blood; Vss: volume of distribution at steady state; t_1/2_: half-life time.

**Table 6 pharmaceutics-15-01282-t006:** LASSBio–1920 tissue partition coefficients in animal simulation module.

Tissue	Kp	Tissue	Kp
Kidney	4.48	Heart	2.59
Adipose	3.97	Brain	2.15
Lung	3.92	Muscle	2.09
Gut	3.41	Bone	1.36
Spleen	3.12	Liver	1.00
Skin	3.10		

**Table 7 pharmaceutics-15-01282-t007:** Absorbed fraction of LASSBio-1920 and controls obtained by the PAMPA-GIT technique.

Compound	Fa (Literature)	Pe (Experimental)	Fa% *
Atenolol	21	0.1	8.62
Acyclovir	52	0.1	8.62
Ceftriaxone	1	0.3	23.70
Coumarin	100	26	100.0
Diclofenac	100	14.4	100.0
Hydrocortisone	91	6.1	99.59
Norfloxacin	35	0.8	51.39
Ranitidine	55	0.2	16.50
Sulfasalazine	12	0.1	8.62
Verapamil	98	7.3	99.86
LASSBio-1920	-	6.6	99.74

* 0–30% = Low absorption/30–69% = Medium absorption/70–100% = High absorption.

**Table 8 pharmaceutics-15-01282-t008:** Permeability coefficient (Pe *) of LASSBio-1920 and controls obtained by PAMPA-BBB technique.

Compound	Pe (Theoretical)	Pe1 (Experimental)	Pe2 (Experimental)	Pe (Experimental Mean)	CNS Permeability **
Atenolol	0.8	0.25	0.39	0.3	NO
Caffeine	1.3	0.36	−0.10	0.1	NO
Enoxacin	0.9	0.48	0.06	0.3	NO
Hydrocortisone	1.9	1.81	0.33	1.1	NO
Ofloxacin	0.8	−0.09	−0.34	−0.2	NO
Testosterone	17	19.58	20.33	20.0	YES
Verapamil	16	17.58	15.12	16.4	YES
LASSBio-1920	-	12.43	10.53	11.48	YES

* Pe = compound permeability coefficient (10–6 cm/s). ** CNS permeability = Absorbed by the Central Nervous System. Note: Values greater than 3.47 mean absorption. Values less than 1.13 mean no absorption. Values in the range 1.13–3.47 are Borderline.

## Data Availability

Not applicable.

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
