# Peer review of "Pharmacokinetic Profile Evaluation of Novel Combretastatin Derivative, LASSBio-1920, as a Promising Colorectal Anticancer Agent"

_pharmaceutics, 2023, doi:10.3390/pharmaceutics15041282_

Round 1

Reviewer 1 Report

Dear Authors
Manuscript describes well that LASSBio-1920 showed promising cytotoxic potential and high selectivity for colon 732 cancer cells (HCT-116) and lung cancer cells overexpressing mutated EGFR (PC-9).
The following steps should be clearer information
1) Please mention up-to-date references in the introduction section.
2) In-Vivo experiment is not done yet, please perform it.

Author Response

I appreciate your comments on the manuscript pharmaceutics-2298729, “Pharmacokinetic profile evaluation of novel combretastatin derivative, LASSBio-1920, as a promising colorectal anticancer agent”.

The observations were very important for the adaptation of this new submitted version. I put in bold the notes that were mentioned.

As for the aspect related to the moderate correction of English, we inform you that the manuscript was reviewed by two native collaborators from the United States, Dr. A. Leyva, who edited the English for submission, and Matthew Little, co-author, who is currently in Brazil as a Fulbright student.

Regarding your Comments and Suggestions to the Authors, I hope to clarify some information:

1) Please mention updated references in the introduction section.

Relevant aspect that you cited was to updating of references in the introduction, which have been reviewed and updated, as listed below.

Updated references:

  • Line 38, Citation 1: Wu, F.; Zhou, Y.; Li, L.; Shen, X.; Chen, G.; Wang, X.; Liang, X.; Tan, M.; Huang, Z. Computational Approaches in Preclinical Studies on Drug Discovery and Development. Front. Chem. 2020, 8, doi:10.3389/fchem.2020.00726.
  • Line 49, Citation 5: Karatoprak, G.Åž.; Küpeli Akkol, E.; Genç, Y.; Bardakcı, H.; Yücel, Ç.; Sobarzo-Sánchez, E. Combretastatins: An Overview of Structure, Probable Mechanisms of Action and Potential Applications. Molecules 2020, 25, 2560, doi:10.3390/molecules25112560.
  • Line 49, Citation 7: Sherbet, G. V. Combretastatin Analogues in Cancer Biology: A Prospective View. J. Cell. Biochem. 2020, 121, 2127–2138, doi:10.1002/jcb.293425.
  • Line 51, Citation 8: Gu, Y.; Ma, J.; Fu, Z.; Xu, Y.; Gao, B.; Yao, J.; Xu, W.; Chu, K.; Chen, J. Development Of Novel Liposome-Encapsulated Com-bretastatin A4 Acylated Derivatives: Prodrug Approach For Improving Antitumor Efficacy. Int. J. Nanomedicine 2019, Volume 14, 8805–8818, doi:10.2147/IJN.S210938.
  • Line 63, Citation 10: Barbosa, G.; Gelves, L.G.V.; Costa, C.M.X.; Franco, L.S.; de Lima, J.A.L.; Aparecida-Silva, C.; Teixeira, J.D.; Mermelstein, C. dos S.; Barreiro, E.J.; Lima, L.M. Discovery of Putative Dual Inhibitor of Tubulin and EGFR by Phenotypic Approach on LASS-Bio-1586 Homologs. Pharmaceuticals 2022, 15, 913, doi:10.3390/ph15080913.

2) In-Vivo experiment is not done yet, please perform it.

Regarding the in vivo experiment, it was not in the initial scope or objective of this article to determine the in vivo antitumor activity. In fact, the submitted manuscript refers to in silico study approaches to molecular interaction and pharmacokinetic prediction. In addition, experiments included in vitro studies of cytotoxicity, cell cycle, target validation (EGFR) and obtaining pharmacokinetic parameters of absorption and metabolism; aiming at future studies in animal models with established data, in order to contribute to the reduction of the number of animals to be used in the next experimental steps.

Reviewer 2 Report

This study focused on investigating the pharmacokinetic properties of LASSBio-1920.  It is important to investigate pharmacokinetics in the development of novel drugs, and although this research examined in detail, I think a little confirmation is necessary before publication.  I recommend the authors to improve the following points.  I hope that my comments are useful for the improvement of the paper.

(1)          In this study, the cytotoxicity and mechanism of action of LASS-62 Bio-1920 were investigated using human colorectal cancer cells (HCT-116) and non-small cell lung cancer cells (PC-9: overexpressed EGFR with L858R mutation).  Please explain why you didn't use other types of cancer cells.

(2)          In Figure 1(B and C) and Figure 2(B-D) and Figure3, cytotoxicity was investigated using different methods, none of which showed dose dependence. How is it considered?  

(3)          Figures 1B and C, Figures 2B and C, respectively, the legend of the figures is reversed.

Author Response

I appreciate your comments on the manuscript pharmaceutics-2298729, “Pharmacokinetic profile evaluation of novel combretastatin derivative, LASSBio-1920, as a promising colorectal anticancer agent”.

The observations were very important for the adaptation of this new submitted version. I put in bold the notes that were mentioned.

Regarding your Comments and Suggestions to the Authors, I hope to clarify some information:

  • In this study, the cytotoxicity and mechanism of action of LASS-62 Bio-1920 were investigated using human colorectal cancer cells (HCT-116) and non-small cell lung cancer cells (PC-9: overexpressed EGFR with L858R mutation).  Please explain why you didn't use other types of cancer cells.

As can be identified in table 1 of the manuscript, we evaluated the cytotoxicity of the sample LASSBio-1920 on 7 different cell lines, such as SF-295, a human glioblastoma cancer cell; PC-3, a human prostate cancer cell line; H-1975, a human lung non-small cell carcinoma cell line; and N-87, a gastric carcinoma.  Thus, based on the CI50 value, we selected for investigation of the molecular mechanism only the human colorectal cancer cells (HCT-116) and non-small cell lung cancer cells (PC-9: overexpressed EGFR with L858R mutation).

  • In Figure 1(B and C) and Figure 2(B-D) and Figure3, cytotoxicity was investigated using different methods, none of which showed dose dependence. How is it considered? 

When selecting the most sensitive strain after testing on a panel of strains of different types, it is recommended that in the next experimental steps, concentrations corresponding to IC50, half and double are used to assess whether the effect has a dose dependent relationship. Regarding the HCT-116 strain, figure 2, the above established was applied and the dose dependent effect was verified in the DNA fragmentation result (Fig. 2D), indicating this concentration range is useful to study the mechanism of action in this strain. Regarding the PC-9 strain (fig 3), preliminary tests were performed in an attempt to define better dose ranges to observe the dose dependent effect. However, as observed in graphs B, C, and D of figure 3, it was not possible to establish this relationship, even using concentrations lower than IC50 of the compound at 48h for this strain.

  • Figures 1B and C, Figures 2B and C, respectively, the legend of the figures is reversed.

In fact, it was the wrong subtitle. It has been corrected as indicated.

Reviewer 3 Report

The manuscript "Pharmacokinetic profile evaluation of novel combretastatin derivative, LASSBio-1920, as a promising colorectal anticancer agent" is an excellent work, describing the in vitro citotoxic activity of N-acylhydrazone derivative LASSBio-1586, LASSBio-1920. The article is well written and organized, giving interesting insights on MOA, molecular docking and pharmacokinetic parameters such as tissue-plasma partition coefficients or metabolic stability. 

Hence, it is suitable for publication after the following minor revisions:

- Introduction must be improved. Add the structure of LASSBio-1920 and its precursors, as described in lines 52-65.

- Furthermore, several literature references are missing. At the end of line 51, cite recent articles about new small molecules developed starting from CA-4: Eur J Med Chem. Rep. 2021, 1, 100004, https://doi.org/10.1016/j.ejmcr.2021.100004; J. Med. Chem. 2016, 59, 7, 3439–3451, https://doi.org/10.1021/acs.jmedchem.6b00101; Eur J Med Chem. 2022 Dec 5;243:114744. doi: 10.1016/j.ejmech.2022.114744. More missing citations at the end of lines 40 and 56.

- Merge figures 6 and 7 to facilitate comparing the results.

- Some typo must be checked. For ex.: correct 105 in line 126; remove t from t0.01 in line 223; check that IC50 is always written correctly.

Author Response

I appreciate your comments on the manuscript pharmaceutics-2298729, “Pharmacokinetic profile evaluation of novel combretastatin derivative, LASSBio-1920, as a promising colorectal anticancer agent”.

The observations were very important for the adaptation of this new submitted version. I put in bold the notes that were mentioned.

Regarding your Comments and Suggestions to the Authors, I hope to clarify some information:

- Introduction must be improved.

Relevant aspect that you cited was to updating of references in the introduction, which have been reviewed and updated, as listed below.

Updated references:

  • Line 38, Citation 1: Wu, F.; Zhou, Y.; Li, L.; Shen, X.; Chen, G.; Wang, X.; Liang, X.; Tan, M.; Huang, Z. Computational Approaches in Preclinical Studies on Drug Discovery and Development. Front. Chem. 2020, 8, doi:10.3389/fchem.2020.00726.
  • Line 49, Citation 5: Karatoprak, G.Åž.; Küpeli Akkol, E.; Genç, Y.; Bardakcı, H.; Yücel, Ç.; Sobarzo-Sánchez, E. Combretastatins: An Overview of Structure, Probable Mechanisms of Action and Potential Applications. Molecules 2020, 25, 2560, doi:10.3390/molecules25112560.
  • Line 49, Citation 7: Sherbet, G. V. Combretastatin Analogues in Cancer Biology: A Prospective View. J. Cell. Biochem. 2020, 121, 2127–2138, doi:10.1002/jcb.293425.
  • Line 51, Citation 8: Gu, Y.; Ma, J.; Fu, Z.; Xu, Y.; Gao, B.; Yao, J.; Xu, W.; Chu, K.; Chen, J. Development Of Novel Liposome-Encapsulated Com-bretastatin A4 Acylated Derivatives: Prodrug Approach For Improving Antitumor Efficacy. Int. J. Nanomedicine 2019, Volume 14, 8805–8818, doi:10.2147/IJN.S210938.
  • Line 63, Citation 10: Barbosa, G.; Gelves, L.G.V.; Costa, C.M.X.; Franco, L.S.; de Lima, J.A.L.; Aparecida-Silva, C.; Teixeira, J.D.; Mermelstein, C. dos S.; Barreiro, E.J.; Lima, L.M. Discovery of Putative Dual Inhibitor of Tubulin and EGFR by Phenotypic Approach on LASS-Bio-1586 Homologs. Pharmaceuticals 2022, 15, 913, doi:10.3390/ph15080913.

- Add the structure of LASSBio-1920 and its precursors, as described in lines 52-65.

It was considered a very interesting suggestion, which allowed comparison of the chemical structures of the compound and the precursors. The figure was elaborated with the aid of MarvinSketch 20.4 software (Chemaxon, Budapest, HUN), placed between lines 51 and 52, with respective reference to line 58 of the new version submitted.

- Merge figures 6 and 7 to facilitate comparing the results.

Your suggestion was accepted

- Some typo must be checked. For ex.: correct 105 in line 126; remove t from t0.01 in line 223; check that IC50 is always written correctly

A new reading was performed on the text and in fact some formatting mistakes were found, which were adjusted as indicated in this new version submitted.

Round 2

Reviewer 1 Report

Dear Author

Manuscript describes well.